# Simulation Study on the Influence of Gas Mole Fraction and Aqueous Activity under Phase Equilibrium

**Weilong Zhao [1], Hao Wu [1], Jing Wen [2], Xin Guo [1], Yongsheng Zhang [1],* and Ruirui Wang [2]**

[1] Henan Province Engineering Laboratory for Eco-architecture and the Built Environment, Henan Polytechnic University, Jiaozuo 454000, China; zhaowl@hpu.edu.cn (W.Z.); 211607020050@home.hpu.edu.cn (H.W.); 211607010023@home.hpu.edu.cn (X.G.)

[2] School of Mechanical and Power Engineering, Henan Polytechnic University, Jiaozuo 454000, China; 211705010028@home.hpu.edu.cn (J.W.); jxndwrr@hpu.edu.cn (R.W.)

* Correspondence: zhangyongsheng@hpu.edu.cn; Tel.:+86-183-368-63941

**Abstract:** This work explored the influence of gas mole fraction and activity in aqueous phase while predicting phase equilibrium conditions. In pure gas systems, such as $CH_4$, $CO_2$, $N_2$ and $O_2$, the gas mole fraction in aqueous phase as one of phase equilibrium conditions was proposed, and a simplified correlation of the gas mole fraction was established. The gas mole fraction threshold maintaining three-phase equilibrium was obtained by phase equilibrium data regression. The UNIFAC model, the predictive Soave-Redlich-Kwong equation and the Chen-Guo model were used to calculate aqueous phase activity, the fugacity of gas and hydrate phase, respectively. It showed that the predicted phase equilibrium pressures are in good agreement with published phase equilibrium experiment data, and the percentage of Absolute Average Deviation Pressures are given. The water activity, gas mole fraction in aqueous phase and the fugacity coefficient in vapor phase are discussed.

**Keywords:** gas mole fraction; activity; UNIFAC; phase equilibrium; threshold value

## 1. Introduction

Gas hydrate is a non-stoichiometric crystalline compound, which consists of a lattice formed by hydrogen bonds of water molecules as the host under high pressure and low temperature. Some gas molecules, such as methane, nitrogen, carbon dioxide and propane, are as the guest firmly surrounded by the crystal network formed by hydrogen bond of water molecules. The ice-like structure enables and stabilizes the existence of gas hydrates at higher temperatures and elevated pressures. The guest molecules must be of correct size to fit inside and stabilize the crystal lattice via weak van der Waals forces with the host water molecules [1,2].

Gas hydrate technology can be applied in many applications, such as gas storage and transportation [3], gas separation [4], refrigeration [5,6], etc. Meanwhile, in response to increasing carbon emissions, the hydrate-based gas separation (HBGS) has attracted the interest of researchers as an effective $CO_2$ capture and storage (CCS) technology [7]. In the last couple of years, vast quantities of natural gas hydrate in the permafrost and deep seabed was found, which is twice as much as the amount of the other fossil fuels combined under a conservative estimate [8]; it makes natural gas hydrate as a kind of potential energy possible. However, gas hydrates can block oil and gas pipelines with high pressure and low temperature inside subsea oil and gas flow line [9]. Furthermore, the methane trapped in gas hydrates is a potent greenhouse gas [10]. In order to solve these problems, scholars conducted a lot of studies and found that adding thermodynamic inhibitors can effectively change the conditions of hydrate formation into higher pressure and lower temperature.

On the contrary, the formation conditions of hydrate can be changed to lower pressure and higher temperature by adding thermodynamic promoters. Regardless of whether inhibiting or promoting hydrate formation, it is necessary to predict phase equilibrium conditions for the above-mentioned applications, and it is important to use reliable and accurate predictive models for predicting hydrate phase equilibria [11].

Among the gas hydrate predictive models of three-phase equilibrium, there are two common thermodynamic prediction models for calculating the phase equilibrium conditions. One is the classical statistical thermodynamic model of van der Waals and Platteeuw [12]. Some improved models designed for the phase equilibrium conditions in distilled water and aqueous solutions were proposed by Nasrifar et al. [13], Haghighi et al. [14], and Martin and Peters [15]. Another hydrate model is the Chen-Guo model [16,17], based on equality of the fugacity in the hydrate and vapor phases, which assumed the activity of water to be in unity and neglected the influence of gas solubility in aqueous phase. However, the changes in water activity caused by the solubility of gases, especially acid gases (e.g., carbon dioxide and hydrogen sulfide), and the addition of inhibitors/promoters should not be ignored [18,19]. Therefore, a better precision activity of water in aqueous phase is the key to improving the Chen-Guo model. Ma et al. [20,21] used the Patel–Teja equation of state (PT EoS), coupled with the Kurihara mixing rule and the one-fluid mixing rule to calculate the water activity in aqueous phase. Sun and Chen [18] combined the modified method introducing the Debye–Hückel electrostatic contribution term with the PT EoS to predict the nonideality of aqueous phase including ionic components. Liu et al. [22] built a simple correlation to calculate the activity of water in methanol–water solutions of sour gases ($CH_4/CO_2/H_2S/N_2$), in which parameters were determined from the phase equilibrium data. Moreover, among the models of aqueous phase activity, the UNIQUAC model [23] and the modified UNIFAC model [24] were constantly used to calculate aqueous phase. Delavar and Haghtalab [25,26] used the Chen-Guo and UNIQUAC models, referring the Soave-Redlich-Kwong-Huron-Vidal equation of state (SRK-HV EoS) conjunction with the Henry's law, to calculate the gas hydrate formation conditions. Dehaghani and Karami [27] employed the predictive Soave-Redlich-Kwong equation of state (PSRK-EoS) along with the modified Huron-Vidal (MHV1) missing rule and UNIQUAC model to calculate fugacity and activity coefficient of water in equilibrated fluid phases. Klauda and Sander [19,28] applied the modified UNIFAC model and PSRK-EoS coupled with the classical mixing rules, and the results obtained were in good agreement with the experimental data.

However, regardless of using UNIFAC or UNIQUAC model to calculate aqueous phase activity, the gas mole fraction in aqueous phase must be obtained first. Thus, in previous literature [19,25–28], Henry's law was used to describe the gas solubility in aqueous phase, and the gas mole fraction in aqueous phase relied on simultaneous Henry constants, the partial molar volume at infinite dilute (presented by Heidemann and Prausnitz [29]) and vapor phase fugacity calculated by the equation of state. It should be noted that both Henry's law and the partial molar volume at infinite dilution are defined on the basis of an imaginary ideal system. Furthermore, some parameters used in calculating the Henry's law constants and the partial molar volume at infinite dilution are also obtained by regression under the assumed ideal state. As a result, when acid gases exist in the actual system, there is a deviation in the water activity and the gas mole fraction in aqueous phase. Therefore, the gas mole fraction in aqueous phase as a function of temperature and pressure is considered one of the factors that influence the phase equilibrium conditions in this work.

In order to minimize the adverse impacts of the Henry's law constants and infinite dilution partial molar volumes on the hydrate equilibrium conditions prediction, we fitted a correlation of gas mole fraction in aqueous phase according to experimental data. Moreover, the UNIFAC model [30–32] and the correlation proposed in this work were employed to calculate the activity coefficient of aqueous phase; PSRK [33–35] was used to calculate vapor phase fugacity, and the Chen-Guo model was used to calculate the fugacity of the hydrate phase. These models not only alleviate empirically fitting the intermolecular parameters required in the van der Waals and Platteeuw model but also avoid the

calculation error of water activity caused by Henry's law and the infinite dilution partial molar volume. It is noteworthy that the framework proposed in this work only applied in the pure gas (such as $CH_4$, $CO_2$, $N_2$ or $O_2$) and distilled water system; the mixed gas system and the additive system will be further studied in future work. Finally, the results calculated are compared with the experimental data in literatures, and the calculated fugacity coefficient of vapor phase and water activity are given.

## 2. Thermodynamic Framework

To predict the phase equilibrium conditions of gas hydrate, the iso-fugacity rule is used in the three phases (vapor, water, and hydrate) and a fugacity approach is considered for both gas and water as:

$$f_i^H(z, T, P) = f_i^L(x, T, P) = f_i^V(y, T, P) \tag{1}$$

where $f_i^H$, $f_i^L$ and $f_i^V$ are the fugacity of component $i$ including water in the hydrate, liquid and vapor phases, respectively; $z$, $x$ and $y$ represent the mole fraction of component $i$ in the hydrate, liquid and vapor phases, respectively. The fugacity of water in hydrate phase, $f_w^H$, is expressed as:

$$f_w^H(T, P) = f_w^{MT}(T) \times exp\left(\frac{-\Delta\mu_w^{MT-L}}{RT}\right) \tag{2}$$

where $f_w^{MT}$ represents the fugacity of water in the hypothetical empty hydrate lattice and is assumed equal to the saturated vapor pressure of the empty hydrate lattice [36]; $-\Delta\mu_{L\ w}^{MT-}$ is the chemical potential difference calculated by the method of Holder et al. [37]; and $R$ is the universal gas constant.

### 2.1. Thermodynamic Model of Vapor Phase

The PSRK group-contribution method is based on the SRK equation of state [38], which is used to calculate the fugacity of components in vapor phase, as:

$$P = \frac{RT}{v_m - b} - \frac{a}{v_m(v_m - b)} \tag{3}$$

where $P$ and $T$ are the system pressure and temperature, respectively; $v_m$ represents the mole volume, which is obtained by solving the cubic equation derived from Equation (3), and the value is the same as the largest real root of the equation [35]; $a$ and $b$ are parameters of PSRK.

The parameters $a_i$ and $b_i$ of pure component $i$ can be calculated from the critical properties $T_{c,i}$ and $P_{c,i}$.

$$a_i = \frac{0.42748R^2T_{c,i}^2}{P_{c,i}}f(T) \tag{4a}$$

$$b_i = \frac{0.08664RT_{c,i}}{P_{c,i}} \tag{4b}$$

$$f(T) = \left[1 + c_1\left(1 - T_r^{0.5}\right)\right]^2 \tag{5}$$

where $T_r = T/T_c$; the pure fluid parameter $c_1$ is taken from the study of Holderbaum and Gmehling [35]. The PSRK mixing rule is written as:

$$a = b\left[\frac{RT\sum y_i ln\gamma_i}{A_1} + \sum y_i\frac{a_i}{b_i} + \frac{RT}{A_1}\sum y_i ln\frac{b}{b_i}\right] \tag{6}$$

$$b = \sum y_i b_i \tag{7}$$

where $\gamma_i$ stands for the activity coefficient of component $i$ calculated by UNIFAC model; the recommended value of $A_1 = -0.64663$ in PSRK model. The activity coefficient $\gamma_i$ is a correction factor

that accounts for deviations of real systems from that of an ideal solution, which can be estimated from chemical models (such as UNIFAC). Thus, the fugacity coefficient is given by:

$$ln\ \varphi_i = \frac{b_i}{b}(z-1) - ln\left[z\left(1-\frac{b}{v_m}\right)\right] - \sigma\ ln(1+\frac{b}{v_m}) \tag{8}$$

$$\sigma = \frac{1}{A_1}\left(ln\gamma_i + ln\frac{b}{b_i} + \frac{b_i}{b} - 1\right) + \frac{a_i}{b_i RT} \tag{9}$$

where $\varphi_i$ is the fugacity coefficient of component $i$; $z = Pv_m/RT$.

### 2.2. Thermodynamic Model of Hydrate Phase

Chen and Guo [16,17] proposed a two-step hydrate formation mechanism for gas hydrate formation. The following two processes are considered simultaneously in the nucleation process of hydrate: (1) A quasi-chemical reaction process to form basic hydrate, and (2) an adsorption process of smaller gas molecules in the linked cavities of basic hydrate, which leads to the non-stoichiometric property of hydrate. The model is expressed as:

$$f_i^H = z_i f_i^0 \left(1 - \sum_j \theta_j\right)^\alpha \tag{10}$$

where $z_i$ denotes the mole fraction of the basic hydrate formed by gas component $i$, and $z_i = 1$ for pure gas; $\theta_j$ represents the fraction of the linked cavities occupied by the gas component $j$; $\alpha$ is the ratio of linked cavities and basic cavities [39], which equals 1/3 for sI hydrates and 2 for sII hydrates, respectively.

$$\sum_j \theta_j = \frac{\sum_j f_j c_j}{1 + \sum_j f_j c_j} \tag{11}$$

where $f_j$ denotes the fugacity of component $j$ in vapor phase calculated by PSRK method; $c_j$ stands for the rigorous Langmuir constant, which is calculated from the Lennard-Jones potential function.

In Equation (10), $f_i^0$ represents the fugacity of component $i$ in vapor phase in equilibrium with the unfilled pure basic hydrate $i$ ($\sum\theta_j = 0$) [21]. According to the Chen-Guo model, it can be calculated as:

$$f_i^0 = exp\left(\frac{-\sum_j A_{ij}\theta_j}{T}\right)\left[A_i' exp\left(\frac{B_i'}{T-C_i'}\right)\right] exp\left(\frac{\beta P}{T}\right) a_w^{\frac{-1}{\lambda_2}} \tag{12}$$

where $A_{ij}$ is the binary interaction coefficient which stands for the interplays between gas molecule $i$ in the basic hydrate and gas molecule $j$ in the linked cavities. $A_i'$, $B_i'$ and $C_i'$ are the Antoine constants, as reported by Chen and Guo [17]. $\beta$ is the function of water volume difference between that in the unfilled basic hydrate phase and the water phase, and the large cavity number per water molecule [20], $\beta = 0.4242$ K/bar for sI hydrates, $\beta = 1.0224$ K/bar for sII hydrates. $a_w$ is the activity of water in aqueous phase, which is calculated by the UNIFAC method. For sI and sII hydrates, $\lambda_2 = 3/23$ and $\lambda_2 = 1/17$, respectively.

### 2.3. Thermodynamic Model of Aqueous Phase

In order to calculate the activity coefficient of components in aqueous phase, the mole fraction of each component in aqueous phase should be obtained first. Therefore, gas is also considered to be a component in aqueous phase. The pressure-corrected Henry's law is employed to calculate the mole fraction of gas component in aqueous phase exclude water as:

$$f_i^L(x_i, T, P) = x_i^L H_i exp\left(\frac{P\overline{V}_i^\infty}{RT}\right) \tag{13}$$

where subscript $i$ represents the gas component in aqueous phase; $H_i$ is the Henry's constant of component $i$, given by the Krichevsky-Kasarnovsky correlation [36,40]; $\overline{V_i}^{\infty}$ is the infinite partial molar volume of the component $i$ in water, given by Heidemann and Prausnitz [29].

With the phase equilibrium, the gas mole fraction in aqueous phase can be calculated by the correlation as:

$$x_i^L = \frac{f_i^V(y_i, T, P)}{H_i exp(\frac{P\overline{V_i}^{\infty}}{RT})} \tag{14}$$

Considering the presence of the additive, whether it is a promoter that raises the phase equilibrium temperature (pressure) or an inhibitor that lowers the temperature (pressure), the components in the liquid phase should be recalculated. Furthermore, Delavar and Haghtalab [25,26] point out that the mole fraction of each component in aqueous phases can be calculated as:

$$n_i = \frac{x_i^L(1 - wt\%)}{M_i} \tag{15}$$

$$n_p = \frac{wt\%}{M_p} \tag{16}$$

where $i$ represents water and gas component; $p$ represents the promoter (inhibitor); $M_i$ and $M_p$ are the molecular weight of component $i$ and the promoter (inhibitor); $wt\%$ stands for the weight percentage of the promoter (inhibitor) in aqueous phase.

$$x_i = \frac{n_i}{\sum n_i} \tag{17}$$

where $i$ represents water, gas components and the promoter (inhibitor); $n_i$ represents the mole fraction of water, gas component and the promoter (inhibitor) in aqueous solutions of a unit mass.

The activity coefficient of the components in aqueous phase is calculated by UNIFAC model [30–32], which consisting of the combinatorial and residual terms, as:

$$ln\gamma_i = ln\gamma_i^C + ln\gamma_i^R \tag{18}$$

where $\gamma_i^C$ and $\gamma_i^R$ stand for the combinatorial term and residual term of component $i$, respectively. The combinatorial term takes the different sizes and shapes of the molecules into account.

$$ln\gamma_i^C = ln\frac{\psi_i}{x_i} + 1 - \frac{\psi_i}{x_i} - \frac{1}{2}Zq_i(ln\frac{\varphi_i}{\theta_i} + 1 - \frac{\varphi_i}{\theta_i}) \tag{19}$$

$$\psi_i = \frac{x_i r_i^{\frac{2}{3}}}{\sum_j x_j r_j^{\frac{2}{3}}} \tag{20a}$$

$$\varphi_i = \frac{x_i r_i}{\sum_j x_j r_j} \tag{20b}$$

$$\theta_i = \frac{x_i q_i}{\sum_j x_j q_j} \tag{20c}$$

where $Z = 10$; $j$ represents all components in aqueous phase; $\varphi_i$ and $\theta_i$ are the volume and surface area fraction of component $i$, respectively; $r_i$ and $q_i$ are the volume and surface area parameters of component $i$, respectively. They are calculated by the van der Waals volumes $R_k$ and surface areas $Q_k$ of the individual group $k$ using equations as follows:

$$r_i = \sum_k V_k^{(i)} R_k \tag{21a}$$

$$q_i = \sum_k V_k^{(i)} Q_k \tag{21b}$$

where $V_k^{(i)}$ is the number of group $k$ in component $i$. The volume $R_k$ parameters and surface areas parameters $Q_k$ of group $k$ are listed in Table 1.

**Table 1.** The UNIFAC group volume and surface-area parameters.

| Main Group | Sub Group | Number | $R_k$ | $Q_k$ |
|:---:|:---:|:---:|:---:|:---:|
| $H_2O$ | $H_2O$ | 4 | 1.506 | 1.732 |
| $CO_2$ | $CO_2$ | 56 | 2.592 | 2.522 |
| $CH_4$ | $CH_4$ | 57 | 2.244 | 2.312 |
| $O_2$ | $O_2$ | 58 | 1.764 | 1.910 |
| $N_2$ | $N_2$ | 60 | 1.868 | 1.970 |

The residual term of component $i$ in Equation (18) is replaced by the solution-of-groups concept [32] as:

$$ln\gamma_i^R = \sum_k V_k^{(i)} (ln\Gamma_k - ln\Gamma_k^{(i)}) \tag{22}$$

where $k$ represents all groups in aqueous phase, including water; $ln\Gamma_k$ stands for the residual activity coefficient of functional group $k$; $ln\Gamma_k^{(i)}$ is the residual activity coefficient of group $k$ in the reference solution containing only component $i$. Both $ln\Gamma_k$ and $ln\Gamma_k^{(i)}$ are calculated as:

$$ln\Gamma_k = Q_k \left[ 1 - ln \left( \sum_m \theta_m \Psi_{mk} \right) - \sum_m \frac{\theta_m \Psi_{km}}{\sum_n \theta_n \Psi_{nm}} \right] \tag{23}$$

$$\theta_m = \frac{Q_m X_m}{\sum_n Q_n X_n} \tag{24}$$

$$X_m = \frac{\sum_m V_m^{(i)} x_i}{\sum_j \sum_k V_k^{(j)} x_j} \tag{25}$$

where $m$ and $n$ are the summations cover different groups of all components in aqueous phase; $i$ in Equation (25) is the same as component $i$ in Equation (22); $\theta_m$ and $X_m$ are the surface area fraction and the mole fraction of group $m$ in the mixture, respectively. The group interaction parameter $\Psi_{nm}$ proposed by Sander et al. [30] is described as:

$$\Psi_{nm} = exp \left( -\frac{u_{nm} - u_{mm}}{T} \right) \tag{26}$$

where $u_{nm}$ and $u_{mm}$ are the adjustable group interaction parameters (energy parameters). For each group–group interaction, the two parameters have the relation of $u_{nm} = u_{mn}$. The gas–gas group interaction-energy parameters $u_{nm}$ and temperature range are given in Table 2.

**Table 2.** Gas–gas group interaction-energy parameters $u_{nm}$ and temperature range.

| Gas | Temperature Range (K) | Gas (56, 57, 58, 60) [a] |
|:---:|:---:|:---:|
| $CO_2$ | 280–475 | 84.2 |
| $CH_4$ | 275–375 | −80 |
| $O_2$ | 250–330 | −260 |
| $N_2$ | 210–330 | −250 |

[a] 56, 57, 58 and 60 are the group numbers of $CO_2$, $CH_4$, $O_2$ and $N_2$ in the UNIFAC group parameter list, respectively.

In order to properly describe the temperature dependence gas solubility, the correlation proposed by Sander et al. [30] has been used as follows:

$$u_{gas-water} = u_0 + \frac{u_1}{\left(\frac{T}{K}\right)}$$

(27)

where $u_0$ and $u_1$ are temperature-independent parameters, shown in Table 3.

**Table 3.** Constants for the calculation of gas–water interaction-energy parameters in the temperature range 273–348K.

| Gas | $u_0$ | $u_1(\times 10^{-5})$ |
|:---:|:---:|:---:|
| $CO_2$ | 980.1 | −1.6895 |
| $CH_4$ | 1059.8 | −2.3172 |
| $O_2$ | 1259.9 | −3.0295 |
| $N_2$ | 1260.4 | −2.7416 |

## 3. Calculation Procedure

The equations described above were solved using codes generated by MATLAB 2014b (MathWork, Beijing, China). The calculation procedure to obtain the phase equilibrium conditions of gas hydrates is summarized in the schematic flow diagram shown in Figure 1.

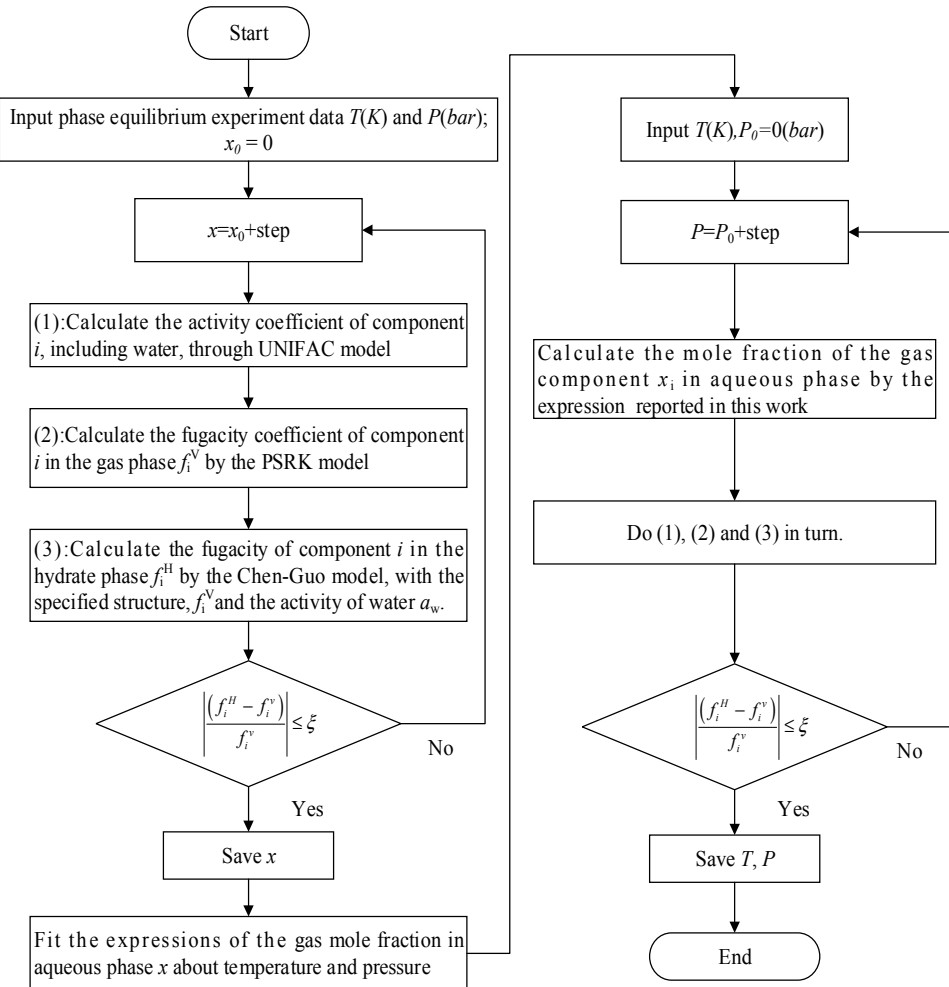

**Figure 1.** Calculation procedure for the prediction of phase equilibrium pressures at given temperatures.

The percentage of the Average Absolute Deviation in Pressure (AADP) is calculated as:

$$AADP(\%) = 100 \sum_{i=1}^{N} \left| \frac{P_i^{\exp} - P_i^{cal}}{P_i^{\exp}} \right| / N \tag{28}$$

where $N$ is the number of experimental points; $P_i^{exp}$ and $P_i^{cal}$ stand for the experimental and calculated pressure, respectively.

## 4. Results and Discussion

When using the UNIFAC model to calculate the activity of gas components, the mole gas fraction in aqueous phase needs to be obtained first. Thus, the Henryconstant $H_i$ and the infinitely diluted partial molar volume $\overline{V_i}^{\infty}$ are employed to calculate the gas mole fraction in aqueous phase. Heidemann and Prausnitz [29] provided a correlation for solving $\overline{V_i}^{\infty}$ as follows:

$$\frac{P_{c,i} \overline{V_i}^{\infty}}{RT_{c,i}} = 0.095 + 2.35 \left( \frac{TP_{c,i}}{c_{11} T_{c,i}} \right) \tag{29}$$

$$c_{11} = \frac{(h^0 - h^s - P_w^s v_w^s + RT)}{v_w^s} \tag{30}$$

where $c_{11}$ represents the cohesive energy for water, which was evaluated at each temperature from thermodynamic properties tabulated; $h^0$ is the molar enthalpy at the given temperature but at zero pressure, and $v_w^s$ is the molar volume of the saturated liquid [29].

However, it is not convenient to get the infinitely diluted partial molar volume of gas in the actual system by the parameters regressed from the assumed ideal state system, especially in the industrial application. Moreover, the parameters are not available for a liquid of unknown components. Therefore, as one of the factors affecting the formation of gas hydrate, the gas mole fraction in aqueous phase cannot be obtained accurately.

As seen from Equation (14), when gas type was given, the gas mole fraction in aqueous phase is only a function of temperature and pressure in phase equilibrium. Furthermore, the gas mole fraction in aqueous phase should be a fixed value in the three-phase equilibrium, which is related to the phase equilibrium temperature and pressure. Therefore, when the equilibrium temperature and pressure are given, the gas mole fraction in aqueous phase can be found by interval search using the framework mentioned in this work. Remarkably, in the process of numerical calculation, due to the unreasonable search step length and the inevitable error of experimental data, a set of phase equilibrium temperature and pressure may correspond to multiple gas mole fraction values. In this case, the average of these values can be taken as the gas mole fraction in aqueous phase under the phase equilibrium. As a result, the correlation of the gas mole fraction in aqueous phase is fitted by temperature and pressure, which is determined by the experimental data and defined as:

$$x_i = a + b \times T + c \times P \tag{31}$$

where $a$, $b$ and $c$ are constants for gas component $i$ + water system, given in Table 4.

**Table 4.** Parameters for the correlation of the gas mole fraction in aqueous phase.

| Guests | $a$ | $b$ $(\times 10^5)$ | $c$ $(\times 10^7)$ | $N_p$ | R-Square |
|--------|-----|---------------------|---------------------|-------|----------|
| $CH_4$ | 0.01713 | −2.58066 | −24.6488 | 500 | 0.99924 |
| $CO_2$ | 0.00454 | 0.403588 | −49.4532 | 200 | 0.89458 |
| $N_2$ | 0.0226 | −5.50785 | 1.49362 | 409 | 1 |
| $O_2$ | 0.02468 | −5.90016 | 1.33948 | 200 | 0.99999 |

**Table 5.** The phase equilibrium pressure and temperature range of experimental data and the Average Absolute Deviation in Pressure (AADP) for predicted results.

| Guests | Temperature (K) | P-Range (bar) | $N_p$ | Reference | AADP (%) |
|---|---|---|---|---|---|
| CH$_4$ | 273.27–289.9 | 26.33–159.52 | 66 | [41] | 1.0634 |
| | 276.81–281.3 | 37.79–62.02 | 5 | This work | 1.6317 |
| CO$_2$ | 270.15–283.15 | 10.19–45.05 | 41 | [42] | 1.5911 |
| N$_2$ | 273.15–291.05 | 160.09–958.53 | 34 | [43] | 1.0224 |
| | 274.55–283.05 | 190.93–453.55 | 3 | [44] | 1.4142 |
| | 279.30–284.00 | 303.00–500.00 | 3 | [45] | 1.5697 |
| O$_2$ | 273.15 | 121.50 | 1 | [43] | 1.6786 |
| | 273.78–284.55 | 138.21–441.30 | 4 | [44] | 1.8001 |
| | 286.27–291.18 | 527.13–953.65 | 6 | [46] | 0.9462 |

Although the correlation of the gas mole fraction in aqueous phase is simple and is multivariate linear form, its precision and calculation accuracy are satisfactory. The number of data points used in fitting Equation (31) and the R-Square are given in Table 4.

As shown in Figure 2, there are two kinds of tendencies for the fugacity coefficient in methane system with the increase of temperature. First, when the temperature is lower than 286.5 K, the fugacity coefficient decreases with the increase of temperature, which has the same tendency with the fugacity coefficient in the carbon dioxide system, although the reduction rate is smaller. On the other hand, when the temperature is higher than 286.5 K, the fugacity coefficient increases exponentially with the temperature increment, which can also be seen in nitrogen and oxygen systems, as displayed in Figure 2. It should be pointed out that the fugacity coefficient decreases with the temperature increment in carbon dioxide system. This indicates that carbon dioxide is more likely to yield to pressure when the temperature is below the critical temperature. Moreover, the exponential growth of the fugacity coefficient in the N$_2$ and O$_2$ systems is mainly related to the temperature increment.

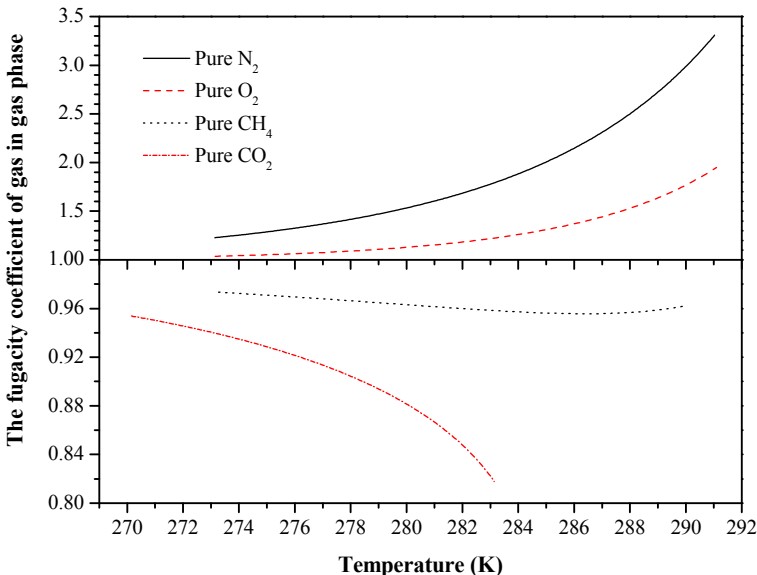

**Figure 2.** The fugacity coefficient of CH$_4$, CO$_2$, N$_2$ and O$_2$ in vapor phase.

The mole fraction of CH$_4$, CO$_2$, N$_2$ and O$_2$ in aqueous phase under phase equilibrium condition is shown in Figures 3–5. In this work, the gas mole fraction was considered as one of the factors affecting the phase equilibrium. It represents the ratio of the number of gas molecules maintaining the three-phase equilibrium to the number of all molecules in aqueous phase.

In Figures 3–5, under phase equilibrium condition, the gas mole fraction in aqueous phase decrease with temperature increment, and all the changing ranges are less than $1 \times 10^{-3}$. Therefore, there may exist a threshold value for the gas mole fraction in aqueous phase. In other words the hydrate will form when the gas mole fraction in aqueous phase reaches a certain threshold value. Furthermore, for methane hydrate, the results in this work are in agreement with the views of Walsh et al. [47] and Guo and Rodger [48]. Walsh et al. suggested that the threshold value of gas mole fraction triggering hydrate formation calculated by the molecular dynamics (MD) simulation was $1.5 \times 10^{-3}$. The threshold value is a reasonable explanation for reducing the temperature or increasing the pressure, which could effectively promote the formation of hydrate. This is because lowering the temperature or increasing the pressure will enhance the gas dissolution, which in turn causes the gas mole fraction in aqueous phase exceeding the threshold value, then hydrate forms.

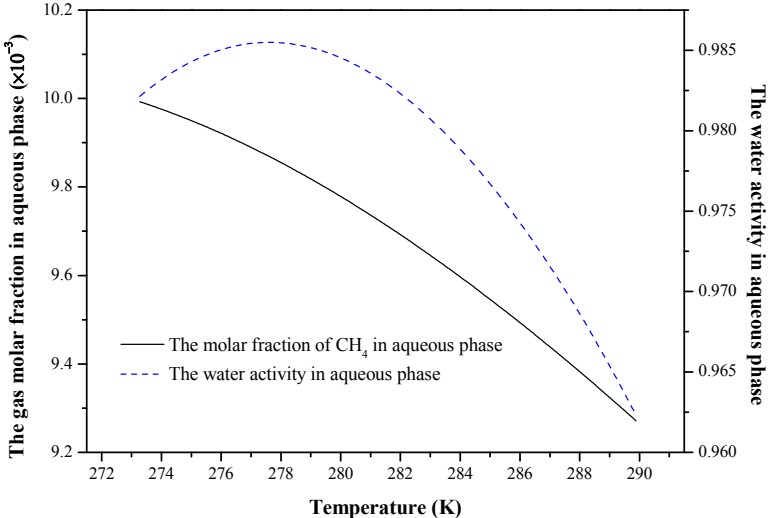

**Figure 3.** The mole fraction of $CH_4$ and water activity in aqueous phase.

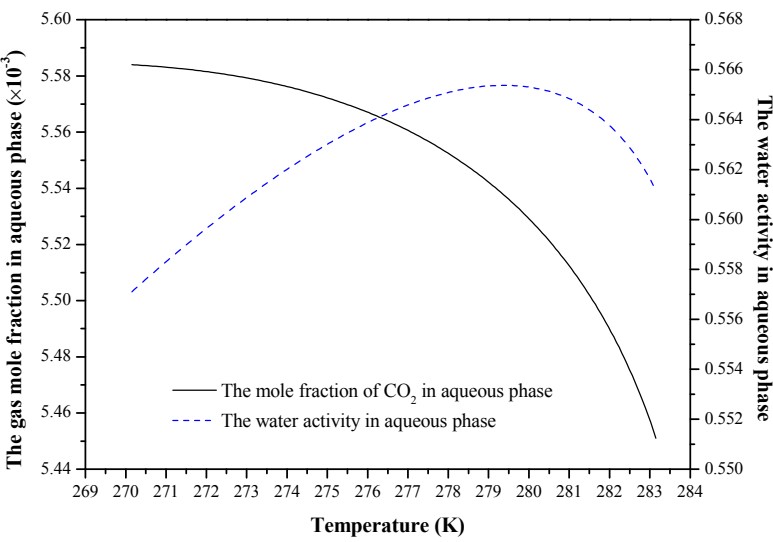

**Figure 4.** The mole fraction of $CO_2$ and water activity in aqueous phase.

In particular, although carbon dioxide has a large solubility in water, the mole fraction of carbon dioxide calculated in this work is not very large under phase equilibrium, as shown in Figure 4. It implied that the total number of carbon dioxide gas molecules in aqueous phase for maintaining the three-phase equilibrium is not large, and it may be much smaller than the sum of the gas molecules dissolved in the water. This is mainly because part of carbon dioxide gas molecules dissolved in water

turn into carbonic acid, thus reducing the amount of carbon dioxide gas molecules existing in water. Meanwhile, the pH of aqueous phase will be changed, and affecting the activity of the water and phase equilibrium conditions.

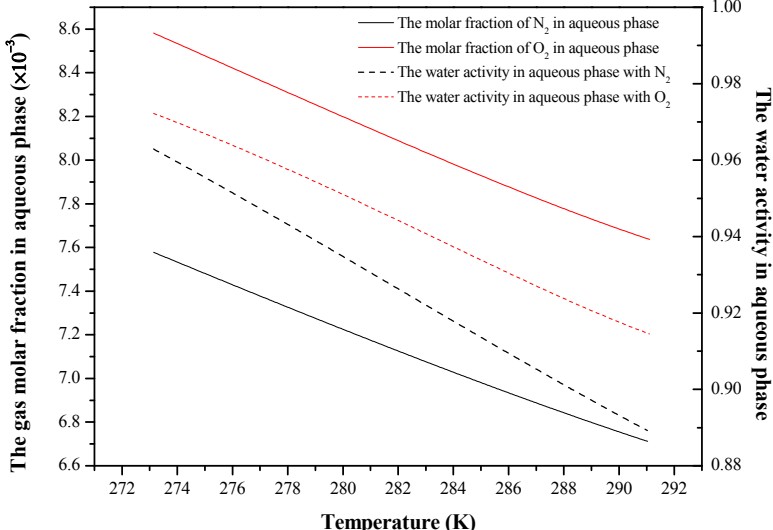

**Figure 5.** The mole fraction of $N_2/O_2$ and water activity in aqueous phase.

In Figure 3, water activity increases with the increase of temperature, in the methane system, reaching a maximum value of 0.985 when the temperature is about 278 K, and then decreases rapidly with the increase of temperature. The general variation trend of the water activity in carbon dioxide system is similar to that of methane system, as seen in Figure 4. In addition, the water activity of the $CO_2$ system reached its maximum value at about 279.5 K. However, the maximum water activity in the $CO_2$ system is only about 0.5658, which is probably because of the effect of the carbonic acid. Nevertheless, the activity of water in aqueous phase decreases almost linearly with temperature increase in nitrogen and oxygen systems, as shown in Figure 5.

Figures 6 and 7 show the experimental and predicted phase equilibrium conditions for the single gas hydrate systems. The temperature range, pressure range and AADP (%) are listed in Table 5.

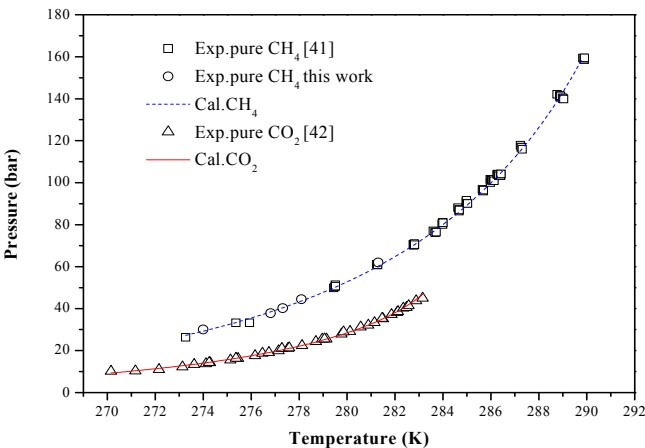

**Figure 6.** Experimental and predicted phase equilibrium conditions for $CH_4/CO_2$ + water systems. Sloan [41], squares □; Ma et al. [42], triangles Δ; this work, circles ○.

Figure 6 shows the experimental and predicted phase equilibrium pressures for $CH_4$ and $CO_2$. It can be seen the predicted results for all the gas systems are in excellent agreement with the experimental data. It should be noted that the type of carbon dioxide hydrate structure was set to sI, and, because

the carbon dioxide gas molecule is too big to be encaged in the linked cavities, the filling rate of the gas molecules in the linked cavities, $\theta_j$, was set to 0, as described by Chen and Guo [16].

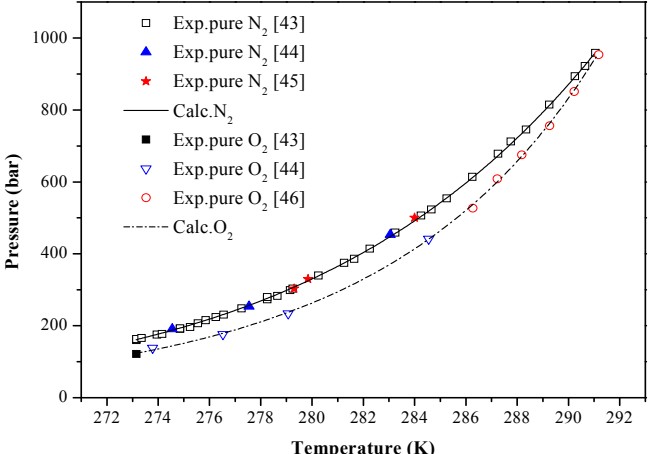

**Figure 7.** Experimental and predicted phase equilibrium conditions for $N_2/O_2$ + water systems. van Cleeff and Diepen [43], squares □, ■; Mohammadi et al. [44], triangles ▲,▽; Duc et al. [45], stars ★; van Cleeff and Diepen [46], circles ○.

The experimental and predicted phase equilibrium pressures for $N_2$ and $O_2$ are displayed in Figure 7. The predicted phase equilibrium pressures are in good agreement with the experiment. It is especially noteworthy that, when calculating oxygen and nitrogen hydrate, the hydrate structure was set to sII, which was based on the ideas proposed by Chen and Guo [16]. This is because the gas molecules of $N_2$ and $O_2$ are small and have a high filling rate in the connected cavities.

The gas mole fraction in Figure 8 was obtained by inverse phase equilibrium data using the framework proposed in this work. The gas mole fraction threshold value for maintaining the three-phase equilibrium state is different to the critical gas concentration. The gas mole fraction threshold value calculated in this work does not contradict the critical gas concentration proposed by Zhang et al. [49]. They pointed out that there is a critical gas concentration in aqueous phase that can spontaneously nucleate in the induction period, and the critical gas concentration is calculated by the total amount of carbon dioxide consumed in vapor phase until hydrate nucleation.

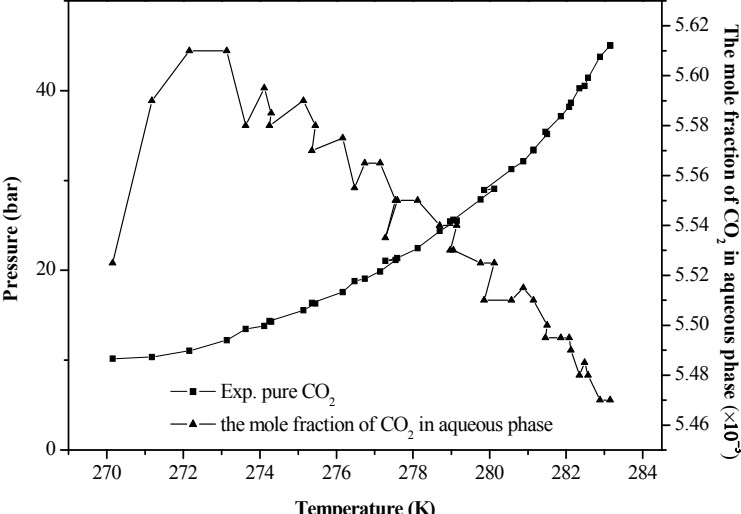

**Figure 8.** The experimental data and the mole fraction of carbon dioxide in aqueous phase for $CO_2$ + water systems under the phase equilibrium. The experimental data were reported by Ma et al. [42].

However, the phase equilibrium data cited in this work were recorded at the end of decomposition rather than in the preliminary stage of nucleation. Gas molecules entrapped in hydrate cage cannot be released totally, which was owing to memory effect [50]. Moreover, there theoretically exists a concentration difference as a force in mass transfer during hydrate nucleation and decomposition. Therefore, the gas mole fraction threshold value calculated in this work is less than the critical gas concentration.

In Figure 8, the threshold value of gas mole fraction achieves a maximum of $5.61 \times 10^{-3}$ at 0 °C. A possible reason is that part of carbon dioxide molecules in aqueous phase react with water to form carbonic acid. When the temperature is above 0 °C and below 0 °C, the pressure increment and the temperature decrement become a dominant factor that results in more stability for the carbonic acid and less solubility of carbon dioxide, respectively. However, this analysis should be proved by further study.

Furthermore, since the correlation of gas mole fraction fitted in this work is a multivariate linear form, the trend of the gas mole fractions in Figures 4 and 8 are different. Therefore, a large number of accurate and reliable experiment data can effectively improve the prediction accuracy of the model in this work.

## 5. Conclusions

In this work, the Chen-Guo model coupled with the PSRK method were employed to predict phase equilibrium conditions of $CH_4$, $CO_2$, $N_2$ or $O_2$ in pure water systems. The gas mole fraction in aqueous phase is one of the factors that affect the phase equilibrium of gas hydrate proposed in this work. The gas mole fraction threshold value maintaining the three-phase equilibrium was obtained by reversed phase equilibrium data. Meanwhile, in order to obtain the water activity in aqueous phase, the correlation of the gas mole fraction threshold value in aqueous phase was fitted though UNIFAC model. The calculated water activity can effectively improve the accuracy of the prediction results, and the predicted results of this work are in good agreement with the experimental data reported in the references.

**Author Contributions:** conceptualization, W.Z. and Y.Z.; methodology, W.Z. and H.W.; software, H.W.; data curation, X.G.; project administration, Y.Z.; resources, Y.Z., writing—original draft preparation, W.Z..; writing—review and editing, J.W. and R.W.

**Funding:** This research was funded by the Young Teacher Capacity Improvement Fund of Henan Polytechnic University, grant number TM2017/02.

**Conflicts of Interest:** The authors declare no conflict of interest.

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
