# Peer review of "Simulation Study on the Influence of Gas Mole Fraction and Aqueous Activity under Phase Equilibrium"

_processes, doi:10.3390/pr7020058_

Round 1

Reviewer 1 Report

There are both water in aqueous activity in Figure 3 and figure 4, why they are not consistent with each other? 

What is “O2+ water” in conclusion?

In introduction “In pure gas systems, such as CH4, CO2, N2 and O2”, should be “or” ?

Figure 8, CO system?

Too many equations, if most of the equations from literature, are all of them necessary in this manuscript?

Should combine table 2 and 3.

Figure 7, symbol for the date is not consistent with the legend and there are three same term in the figure, which make confusions   

Can this model predict the mixture gases of CH4, CO2, N2 and O2?

What is project administration and resources for the contribution for this manuscript?

Author Response

Point 1: There are both water in aqueous activity in Figure 3 and figure 4, why they are not consistent with each other?

Response 1: The water activity reaches a maximum value of 0.985 when the temperature is about 278K in the methane system and a maximum value of 0.5658 at about 279.5K in the carbonic acid system. The difference is most likely because the presence of carbonic acid in the carbon dioxide system affects the activity of water. Under high pressure, the carbon dioxide dissolved in water reacts with water molecules to produce carbonic acid, then the PH of aqueous phase is changed and affects the water activity. This probable reason also shown in lines 273-282 in the manuscript.

Point 2: What is “O2+ water” in conclusion?

Response 2: The “O2+water” represents the gas phase is pure oxygen and the liquid phase is pure water. The related content has been improved in line 333.

Point 3: In introduction “In pure gas systems, such as CH4, CO2, N2 and O2”, should be “or”?

Response 3: We accept the reviewer’s recommend and make correction in the revised manuscript. (line 93)

Point 4: Figure 8, CO system?

Response 4: We are very sorry for the mistake, it is “CO2 system”. This error has been fixed and Figure 8 has been updated.

Point 5: Too many equations, if most of the equations from literature, are all of them necessary in this manuscript?

Response 5: Although there are many equations in this work, they are all necessary for the model. Many parameters in main equations were explained. In addition, because many references were cited in this work and not inconvenient for readers. Therefore, we believe that detail equations are necessary.

Point 6: Should combine table 2 and 3.

Response 6: Through our serious discussion, we think it should keep table 2 and 3 separate. In Table 3, u0 and u1 are temperature-independent parameters. They are constants proposed by Sander et al. ( Sander, B.; Skjold-Jørgensen, S.; Rasmussen, P. Gas solubility calculations. I. Unifac. Fluid Phase Equilibria 1983, 11 (2), 105-126.) for calculating gas-water interaction-energy parameters in the temperature range of 273-348k.

Furthermore, the temperature range of gas-gas group interaction-energy parameters and the constants are different. In order to distinguish between gas-gas group interaction-energy parameters and the constants, we modified the title of Table 3 in the revised manuscript. ( lines 191-192 ).

Point 7: Figure 7, symbol for the date is not consistent with the legend and there are three same term in the figure, which make confusions

Response 7: Three different symbols were used to distinguish experimental data from different literatures. According to comments, we updated the symbol and the legend, and then added the reference number into figure 7.

Point 8: Can this model predict the mixture gases of CH4, CO2, N2 and O2?

Response 8: This model can predict the mixture gases of CH4, CO2, N2 and O2. We offered mixing rules in this model. As described in lines 93-94, the mixed gas system and the additive system are in the process of research now.

Point 9: What is project administration and resources for the contribution for this manuscript?

Response 9: The project administration and resources aim for separating and extracting CH4 in Low concentration coalbed methane (CBM), which is composed of CH4, CO2, N2 and O2. This manuscript could provide a basic prediction model for both pure gas /pure water or mixture gases /mixture liquid systems.

Reviewer 2 Report

In this study, the investigators investigate how best to model three phase equilibrium of gas hydrate systems. This builds upon and improves upon previous studies, where factors are identified as being important that were previously neglected. The modeling of gas hydrate systems is currently an important topic of interest, and believe this work will be of interest to the readers of this journal.

I offer no criticisms of the scientific merit and methods used. Nonetheless, I believe there is much that can be done to improve the quality of the manuscript. By improving the quality of the written document, I believe the impact will be greatly improved. Upon reading the manuscript, I get the perspective that the author is an expert in the field, and has a great depth of knowledge on the subject matter. While the manuscript in its present form I am sure can be well understood by an expert such as the reader, I believe that efforts need to be made to clarify the material for the non-expert. If the interested non-expert can not understand what you are doing, they will be unlikely to adopte (or correctly use) your proposed strategy. I offer several examples that stood out to me, but would encourage the authors to thoroughly re-review the manuscript.

Section 2.1, equation (6). A parameters in the mixing rule is the activity coefficient, that is calculated using UNIFAC. Can you clarify this term? When I think of UNIFAC, I think of computing liquid phase activity coefficients. After all, UNIFAC neglects the pressure dependence of the activity coefficient. However, if this is a gas phase activity coefficient as might be suggested by the section title, how do you account for pressure? Or is it a liquid phase activity coefficient you are computing. Please walk me through the calculation. I know you provided plenty of citations. However, I do not want to look at several manuscripts just to understand this one equation. Not to mention, one of the beauties of this journal is it is freely available, unlike the journals you reference. You those reference may not be available to the interested reader.

Following my last comment, are the mixing rules optimized around the use of UNIFAC to compute the activity coefficient? Guessing by the parameter A, it was optimized are computed the activity coefficients with some model. Otherwise, could they be improved by using a model such as mod-UNIFAC which we expect to do a better job predicting limiting activity coefficients? mod-UNIFAC (and there are many varieties of mod) was mentioned just once in the introduction.

Following equation (3), you mention that that v is the partial molar volume of component i at infinite dilution. Please clarify that this is for component i in water.

Please clarify equations (15) and (16). Where did they come from? What is the role of the inhibitor and promoter, and how to they relate to the origin of these equations?

Equation (20a). You mentioned using UNIFAC. However, in the Flory-Huggins portion of the combinatorial term you scale the van der Waals volume by 2/3 power. This scaling is not used in teh original UNIFAC model but was used in some of the modified versions adopted by others. Please clarify what model you are using.

Secton 3 Calculation Procedure. Can you provide a copy of your MATLAB code as a supporting information document?

Can you discuss where equation (29) comes from and what assumptions it makes?

Equation (30) and table (4). How many data points were used in the correlation? This would be useful to interpret the reported R^2 and the appropriateness of the fit.

Again, these are not criticisms of the science, but I have issues with the clarify of the manuscript.

Author Response

In this study, the investigators investigate how best to model three phase equilibrium of gas hydrate systems. This builds upon and improves upon previous studies, where factors are identified as being important that were previously neglected. The modeling of gas hydrate systems is currently an important topic of interest, and believe this work will be of interest to the readers of this journal.

I offer no criticisms of the scientific merit and methods used. Nonetheless, I believe there is much that can be done to improve the quality of the manuscript. By improving the quality of the written document, I believe the impact will be greatly improved. Upon reading the manuscript, I get the perspective that the author is an expert in the field, and has a great depth of knowledge on the subject matter. While the manuscript in its present form I am sure can be well understood by an expert such as the reader, I believe that efforts need to be made to clarify the material for the non-expert. If the interested non-expert can not understand what you are doing, they will be unlikely to adopte (or correctly use) your proposed strategy. I offer several examples that stood out to me, but would encourage the authors to thoroughly re-review the manuscript.

Point 1: Section 2.1, equation (6). A parameters in the mixing rule is the activity coefficient, that is calculated using UNIFAC. Can you clarify this term? When I think of UNIFAC, I think of computing liquid phase activity coefficients. After all, UNIFAC neglects the pressure dependence of the activity coefficient. However, if this is a gas phase activity coefficient as might be suggested by the section title, how do you account for pressure? Or is it a liquid phase activity coefficient you are computing. Please walk me through the calculation. I know you provided plenty of citations. However, I do not want to look at several manuscripts just to understand this one equation. Not to mention, one of the beauties of this journal is it is freely available, unlike the journals you reference. You those reference may not be available to the interested reader.

Response 1: We accept the useful comments and suggestions from you. The activity coefficient of the components in a system is a correction factor that accounts for deviations of real systems from that of an Ideal solution, which can either be measured via experiment or estimated from chemical models (such as UNIFAC). We clarify this term in lines 118-120 in the revised manuscript.

In equation (6), γi stands for the activity coefficient of component i in liquid phase, which is calculated by equation (18) in Section 2.3 in the revised manuscript. Moreover, T. Holderbaum and J. Gmehling (Holderbaum, T.; Gmehling, J. PSRK: a group contribution equation based on UNIFAC. Fluid Phase Equilibria 1991, 70, 251-265.) pointed out that The mixture parameter a is calculated by the UNIFAC method. Besides, under the phase equilibrium, gas are present in vapor, aqueous, and hydrate phases. In the UNIFAC model, the pressure is indeed neglected, but gas is considered to be a component in liquid phase and the amount of gas dissolved in the liquid phase is related to the pressure of system, which in turn affects the mole fraction of each component in the liquid phase.

Point 2: Following my last comment, are the mixing rules optimized around the use of UNIFAC to compute the activity coefficient? Guessing by the parameter A, it was optimized are computed the activity coefficients with some model. Otherwise, could they be improved by using a model such as mod-UNIFAC which we expect to do a better job predicting limiting activity coefficients? mod-UNIFAC (and there are many varieties of mod) was mentioned just once in the introduction.

Response 2: The model established in this work is mainly for the prediction of the mixing system, which is in the process of research now. Meanwhile, when calculating the activity coefficient using a chemical model, the group assignment method and the number of groups are highly evaluated. Because in the system with additives, a variety of group parameters are needed, and the presence or absence of group parameters (especially which we might use) will greatly affect whether the model can be used.

The original UNIFAC has more published parameters and group assignment that we will need later. The parameters of the original UNIFAC model can be find on http://www.ddbst.com.

Point 3: Following equation (3), you mention that that v is the partial molar volume of component i at infinite dilution. Please clarify that this is for component i in water.

Response 3: In Equation (13), there is a parameter , which represents the infinite partial molar volume of the component i. We accept the useful comments and clarify  in line 150.

Point 4: Please clarify equations (15) and (16). Where did they come from? What is the role of the inhibitor and promoter, and how to they relate to the origin of these equations?

Response 4: If considering the presence of the additive in water, whether it is an promoter that raises the phase equilibrium temperature (pressure) or an inhibitor that lowers the temperature (pressure), the components in the liquid phase should be recalculated. And, Delavar and Haghtalab (Delavar, H.; Haghtalab, A. Prediction of hydrate formation conditions using GE-EOS and UNIQUAC models for pure and mixed-gas systems. Fluid Phase Equilibria 2014, 369, 1-12.) point out that the mole fraction of each component in aqueous phases can be calculated by Eq.(15) and (16). This is also explained in line 153-156 in the revised manuscript.

Point 5: Equation (20a). You mentioned using UNIFAC. However, in the Flory-Huggins portion of the combinatorial term you scale the van der Waals volume by 2/3 power. This scaling is not used in the original UNIFAC model but was used in some of the modified versions adopted by others. Please clarify what model you are using.

Response 5: Equation (20a) was mentioned by Sander et al (30.       Sander, B.; Skjold-Jørgensen, S.; Rasmussen, P. Gas solubility calculations. I. Unifac. Fluid Phase Equilibria 1983, 11 (2), 105-126.) and first proposed by Kikic et al (Kikic, I.; Alessi, P.; Rasmussen, P.; Fredenslund, A. On the combinatorial part of the UNIFAC and UNIQUAC models. The Journal of Chemical Thermodynamics 1980, 58, 253-258.).

Point 6: Secton 3 Calculation Procedure. Can you provide a copy of your MATLAB code as a supporting information document?

Response 6: MATLAB code for CH4 has been uploaded but the notes are written in Chinese.

Point 7: Can you discuss where equation (29) comes from and what assumptions it makes?

Response 7: Equation (29) was provided by Heidemann and Prausnitz (Heidemann, R. A.; Prausnitz, J. M. Equilibrium Data for Wet-Air Oxidation. Water Content and Thermodynamic Properties of Saturated Combustion Gases. Industrial & Engineering Chemistry Process Design & Development 1977, 16 (3), 375-381.), and the original form as follows

(1)

(2)

where c11 represent the cohesive energy for water, which was evaluated at each temperature from thermodynamic properties tabulated; h0 is the molar enthalpy at the given temperature but at zero pressure, and vs w is the molar volume of the saturated liquid.

In the original literature, we did not find any useful assumptions about the equation, and some parameters in this equation is not convenient to get, such as c11, h0 and vs w. Therefore, the gas mole fraction in aqueous phase cannot be obtained accurately. As an alternative, when the equilibrium temperature and pressure are given, the gas mole fraction in aqueous phase can be found by interval search using the framework mentioned in this work. According to your comments, we made some discuss in line 212-222 of the revised manuscript.

Point 8: Equation (30) and table (4). How many data points were used in the correlation? This would be useful to interpret the reported R^2 and the appropriateness of the fit.

Response 8: We appreciate the detailed and useful comments and suggestions from you. There are 500, 200, 409 and 200 data points were used for the correlation parameters of CH4, CO2, N2 and O2, respectively. We added the number of data points used in fitting the equation to Table (4).

Round 2

Reviewer 2 Report

I thank the readers for carefully reviewing the comments I previously raised. The author modified the manuscript as they felt best. While they did not makes changes to the manuscript to address every comment raised, I believe they used their best judgement keeping in mind my comments as whole.